# Reproducibility of Breech Progression Angle: Standardization of Transperineal Measurements and Development of Image-Based Checklist for Quality Control [note 1]

**DOI:** 10.3390/diagnostics15141757

**Published:** 2025-07-11

**Authors:** Ana M. Fidalgo, Adriana Aquise, Francisca S. Molina, Aly Youssef, Otilia González-Vanegas, Elena Brunelli, Ilaria Cataneo, Maria Segata, Marcos J. Cuerva, Valeria Rolle, Maria M. Gil

**Affiliations:** 1Department of Obstetrics and Gynecology, Hospital Universitario de Torrejón, 28850 Madrid, Spain; anamariafidalgoconde@gmail.com (A.M.F.); adriana.aquise@gmail.com (A.A.); 2School of Medicine, Universidad Francisco de Vitoria, Pozuelo de Alarcón, 28223 Madrid, Spain; 3Department of Obstetrics and Gynecology, Hospital Universitario San Cecilio, 18016 Granada, Spain; fsoniamolina@gmail.com (F.S.M.); otigonzalez16@gmail.com (O.G.-V.); 4Instituto de Investigación Biosanitaria ibs.GRANADA, 18012 Granada, Spain; 5Obstetrics Unit, IRCCS Azienda Ospedaliero-Universitaria di Bologna, 40138 Bologna, Italy; aly.youssef78@gmail.com (A.Y.); elenabrunelli23@gmail.com (E.B.); 6UOC Ginecologia ed Ostetricia, Ospedale Maggiore Bologna, 40133 Bologna, Italy; cataneo.ilaria@gmail.com (I.C.); m.segata@ausl.bologna.it (M.S.); 7Department of Obstetrics and Gynecology, Hospital Universitario La Paz, 28046 Madrid, Spain; marxichos@hotmail.com; 8Statistics and Data Management Unit, iMaterna Foundation, Alcalá de Henares, 28806 Madrid, Spain; 9Biostatistics and Epidemiology Platform, Fundación para la Investigación y la Innovación Biosanitaria del Principado de Asturias (FINBA), 33011 Asturias, Spain; 10Vicerrectorado de Investigación, Facultad de Medicina, Universidad Francisco de Vitoria, Carretera Pozuelo a Majadahonda, Km 1.800, Pozuelo de Alarcón, 28223 Madrid, Spain

**Keywords:** breech presentation, reproducibility, transperineal ultrasound, intrapartum ultrasound, cephalic external version

## Abstract

**Highlights:**

**What are the main findings?**
The breech progression angle is a feasible and highly reproducible transperineal parameter when an image-based checklist is used for its evaluation.

**What is the implication of the main finding?**
Image-based checklists can be useful for the standardization of transperineal ultrasound and improve reproducibility of intrapartum parameters.This contributes to increase diagnostic accuracy when performing an External Cephalic Version maneuver or a breech delivery.

**Abstract:**

**Objectives:** To evaluate the reproducibility of measurements of breech progression angle (BPA) by transperineal ultrasound (US) before and after its standardization by applying an image-based checklist. **Methods:** Eighteen 3-dimensional (3D) volumes of transperineal US from women at 36–40 weeks of gestation with a singleton fetus in breech presentation were provided to eight operators from four maternity units in Spain and Italy. All operators measured the BPA using 3D US volume processing software, and interobserver reproducibility was evaluated using the intraclass correlation coefficient (ICC). Following an online live review of all measurements by the operators, and the identification of sources of disagreement, an image-based scoring system for BPA measurement was collaboratively developed. The checklist included the following: (1) acquisition in the midsagittal plane, avoiding the posterior shadow of the pubic ramus; (2) visualization of the complete “almond-shaped” pubic symphysis; (3) drawing a first line along the longitudinal axis of the symphysis, dividing it equally; (4) extending this line to the inferior edge of the bone; and (5) drawing a second line tangentially from the lower edge of the symphysis to the lowest recognizable fetal part. The BPA measurements were then repeated using this checklist, and reproducibility was reassessed. **Results:** Eighteen volumes were analyzed by the eight operators, achieving a moderate reproducibility (ICC: 0.70, 95% confidence interval (CI): 0.48 to 0.86). A score was developed to include a series of landmarks for the appropriate assessment of BPA. Subsequently, the same eighteen volumes were reassessed using the new score, resulting in improved reproducibility (ICC: 0.81, 95% CI: 0.66 to 0.92). **Conclusions:** The measurement of BPA is feasible and reproducible when using a standardized image-based score.

## 1. Introduction

Breech presentation occurs in 3–4% of singleton term pregnancies [1,2]. Over the past decades there has been an increase in elective cesarean sections performed for this reason, driven largely by concerns surrounding the safety of vaginal breech delivery. This shift was particularly catalyzed by the results of the Term Breech Trial, which reported higher rates of neonatal morbidity and mortality associated with planned vaginal birth in breech presentations [3]. As a consequence, there has been a decline in clinical experience with breech deliveries and an associated rise in maternal morbidity and mortality related to cesarean birth [4], together with longer recovery times, and complications in subsequent pregnancies, such us placenta accreta spectrum [4,5]. Therefore, there is a renewed interest in exploring safe strategies to reduce unnecessary cesareans in breech presentations. One such strategy is the promotion of external cephalic version (ECV), a procedure aimed at rotating the fetus into cephalic presentation before the onset of labor [6]. The success of ECV, however, is highly dependent on several factors, including the engagement of the fetal breech within the maternal pelvis [7].

Traditionally, the evaluation of fetal descent and engagement has relied heavily on digital vaginal examination or abdominal palpation. These methods are inherently subjective and can vary significantly between practitioners [8]. In recent years, intrapartum ultrasound (US) has emerged as a valuable tool for providing more objective and reproducible information about fetal position, descent, and engagement [9,10,11,12,13,14,15,16]. Among various US parameters, the angle of progression (AoP) has gained prominence as a reliable measure in cephalic presentations [17,18,19,20,21]. In cases of breech presentation, the breech progression angle (BPA) represents an analogous parameter, providing quantifiable insight into the descent of the breech within the maternal birth canal [21].

The BPA was first described by Youssef et al. in 2021, as the angle between a line running along the long axis of the pubic symphysis and another line extending from the most inferior portion of the pubic symphysis tangentially to the surface of the lowest recognizable fetal part in the maternal pelvis (whether the buttock or the lowest fetal foot), as obtained by transperineal US [21]. The authors reported high reproducibility in a limited single-center study. However, the generalizability of BPA across different clinical settings and among various practitioners has not been extensively evaluated.

The main objective of this study was to assess the reproducibility of the US measurement of the BPA before and after its standardization by applying a newly developed image-based checklist to ensure homogeneity and quality.

## 2. Methods

### 2.1. Study Design and Population

This was a multicenter, observational study carried out between January and May 2023 in four hospitals: Azienda Ospedaliero-Universitaria Policlinico S. Orsola-Malpighi (Bologna) and Ospedale Maggiore di Bologna (Bologna) in Italy, and Hospital Universitario de Torrejón (Madrid) and Hospital Clínico San Cecilio (Granada) in Spain.

The study was approved by the local research ethics committee (Comité de Ética de la Investigación con Medicamentos de los Hospitales Universitarios Torrevieja y Elche-Vinalopó on 29 October 2020 and approved in all participating centers), and written informed consent was obtained from all the participants after providing detailed information about the study.

Twenty transperineal 3-dimensional (3D) US volumes were acquired from pregnant women at 36–40 weeks of gestation with a singleton fetus in breech position. For volume acquisition, participants were assessed in the lithotomy position with empty bladder to ensure consistency in image conditions. The US examinations were performed using either a Voluson SWIFT (GE Healthcare, Zipf, Austria) or Voluson P8 (GE Healthcare, Zipf, Austria) US machine, equipped with a convex transducer covered by a sterile glove. The transducer was positioned in the midsagittal plane visualizing the pubic symphysis, the urethra, the vagina, and the fetal breech or foot as previously described [21].

Following acquisition, the image quality was reviewed by the central study team, and two 3D volumes (volumes 7 and 14) were excluded due to suboptimal visualization, leaving eighteen for final analysis. These were then distributed to eight senior obstetricians, two from each participating unit. To reduce the influence of baseline skill variability, operators were selected among obstetricians with expertise in obstetric US and those who routinely used intrapartum US in their clinical practice but without any previous experience measuring BPA. Although the original publication was reviewed by each operator, no specific training was provided. The operators measured the BPA using a 3D US volume processing software (4D View^TM^, version 5.0, GE Medical, Kretz Ultrasound, Zipf, Austria).

Following the first round of measurements, A.M.F. reviewed all measurements from all operators to identify sources of disagreement, which were subsequently discussed during a live, interactive online session. All investigators collaboratively analyzed the six volumes with the greatest inter-operator variability, as well as those that best illustrated the most common conflicts. The session focused on identifying anatomical landmarks and clarifying interpretation challenges. Based on this discussion, an image-based checklist was developed to standardize the BPA measurement. The checklist included five key criteria: (1) correct midsagittal plane identification, (2) complete delineation of the pubic symphysis, (3) accurate bisecting of the symphysis, (4) precise extension of the first line to the lower edge of the bone, and (5) consistent application of the second line tangent to the fetal breech (Figure 1, Table 1).

Six months later, all operators repeated the BPA measurements using this checklist. This interval was intended to minimize recall bias and assess long-term adherence to standardized procedures.

### 2.2. Statistics

The interobserver reproducibility of the volumes was assessed on two occasions: first before the implementation of the checklist and again after. This allowed us to evaluate the impact of the checklist on improving the consistency of the measurements. The intraclass correlation coefficient (ICC) with 95% confidence intervals (CIs) was used both times to quantify the reproducibility.

Each rater was asked to rate every sample independently, to obtain one measurement from each operator. We computed the ICC2, which, according to the classification of Shrout and Fleiss, is appropriate to accommodate a random sample of judges rating each sample [22]. This approach assumes that both the operators and samples are obtained randomly from their populations. The overall agreement between all operators, as well as paired agreement between every two operators, were estimated. The ICC was categorized as very good (>0.90), good (0.90 to 0.80), moderate (0.80 to 0.60), or poor (<0.60).

Additionally, Bland–Altman plots were generated for each pair of operators to illustrate the mean difference between measurements (bias) and the limits of agreement, along with their 95% CI [23]. The variability in the measurements was also visually assessed by plotting the measurements obtained for each volume.

A post hoc power analysis was conducted to verify that the study was adequately powered to assess reproducibility.

Analyses were performed using the statistical software R (version 4.4.1) [24]. The DescTools package [25] was used to calculate the ICCs, dplyr for data wrangling, table1 for the tables, BlandAltmanLeh for Bland–Altman stats and plots [26], ICC.Sample.Size for the power analysis [27], and the ggplot2 package [28] to generate the plots.

## 3. Results

Prior to the implementation of the checklist, the overall interobserver reproducibility was moderate, with an ICC of 0.70 (95% CI 0.48 to 0.86), and paired agreement between operators varied substantially, ranging from 0.46 to 0.92, with better agreement between operators from the same center, as shown in Table 2. Figure 2 (panel A) represents the individual measurements of each operator for each volume. After applying the checklist, the reassessment of BPA across all volumes by all operators showed a good overall reproducibility, increasing the ICC to 0.81 (95% CI 0.66 to 0.92). Agreement between paired operators also improved, ranging from 0.72 to 0.97, as well as reproducibility between operators from different centers (Table 2 and Figure 2, panel B).

The Bland–Altman plots can be seen in Figure 3 and their statistics in Table 3.

Assuming an expected ICC of 0.8 and a minimum acceptable ICC of 0.4, the post hoc power analysis indicated a statistical power of 79.5%, suggesting that the study was adequately powered to detect acceptable levels of agreement.

Operators reported that the checklist was intuitive and facilitated a more confident interpretation of anatomical landmarks.

## 4. Discussion

### 4.1. Main Findings

This study has demonstrated that the BPA is a reproducible sonographic parameter when measured according to well-standardized criteria. The implementation of the image-based checklist developed for this study improved interobserver reproducibility, both within individual centers and across different institutions. These findings reinforce the importance of structured training and methodological standardization in US assessment, especially when introducing new sonographic parameters into clinical practice.

### 4.2. Comparison with Previous Studies

The original study by Youssef et al., first introducing the BPA, reported good inter-operator reproducibility, with an ICC of 0.83 (95% CI, 0.71–0.90) [21]. However, that study was limited to only two operators from a single center. In contrast, our study involved eight operators from four different institutions, introducing variability that better represents real-world clinical settings. In our study, the maternal pubic symphysis and fetal parts were always identifiable, allowing BPA measurements by all operators in 90% (18 out of 20) of the volumes. Nevertheless, in our first round of measurements, conducted after critically reviewing the original article on a personal basis, the overall ICC was only 0.70 (95% CI 0.48 to 0.86). Notably, agreement tended to be higher among operators from the same center despite blinding, likely due to shared interpretations and informal discussion of the original methodology. Similar limitations in interobserver reproducibility have been observed in other studies [29]. After the introduction of the image-based checklist and structured consensus-building, reproducibility improved markedly, with an ICC of 0.81 (95% CI, 0.66–0.92), consistent with the findings of Youssef et al. These results support the conclusion that the BPA is a feasible parameter for clinical application, provided it is used within a standardized framework and guided by clear protocols.

Operator experience is another well-known factor influencing reproducibility. Duckelman et al. investigated whether clinical US experience affected the reliability of the AoP measurements [18]. In their study, 44 images were acquired by a senior obstetrician with over 10 years of experience from women during their second stage of labor. The images were later analyzed offline, and the AoP was measured by nine examiners with varying levels of US experience: three senior obstetricians (>10 years of experience), three fellows (<five years of experience), and three midwives with no prior US experience in US), using standardized criteria. The overall ICC for the nine operators was 0.72 (95% CI 0.63 to 0.81). Although greater experience was associated with higher ICC values, the difference was not statistically significant. They concluded that while US experience influences performance in the measurement of the AoP, standardized criteria can mitigate this effect. Similarly, Sainz et al. reported significant variability in subjective intrapartum ultrasound assessments, even among experienced operators [29]. In our study, all operators were senior obstetricians with more than five years of ultrasound experience. The improvement in interobserver agreement following implementation of the image-based score highlights the importance of standardized methodology regardless of operator expertise.

Other sonographic parameters for assessing breech progression and predicting delivery outcome have been explored. Jennewein et al. conducted a prospective study showing that sonographic confirmation of fetal breech engagement (termed the intrapartum ultrasound breech engagement sign, or IPUBES) was associated with a significantly lower rate of cesarean delivery. However, because this sign relies solely on qualitative interpretation of sagittal ultrasound images depicting the maternal pubic bone and fetal buttocks, it remains subjective and potentially limits its reproducibility and clinical utility [30]. In contrast, our study contributes to the development of measurable, objective ultrasound parameters, such as the BPA, enabling consistent interobserver comparisons and offering a practical framework for use across diverse clinical settings.

Similarly, other ultrasound parameters have also been investigated to assess the likelihood of success in ECV. Among these, the volume of amniotic fluid is one of the most extensively studied. Noteworthy is the measurement of the volume of amniotic fluid beneath the presenting fetal part (fore-bag), which serves as an indirect but meaningful predictor. Isakov et al. demonstrated that a larger fore-bag significantly increased the probability of successful ECV, incorporating this measurement into a multivariable predictive model with high discriminative capacity (C-index = 0.933) alongside maternal body mass index and parity [31]. Other studies have also emphasized the importance of overall amniotic fluid volume in predicting ECV outcomes [32,33,34]. Despite its utility, the assessment of amniotic fluid volume is subject to certain limitations, particularly concerning reproducibility. Although interobserver variability in ultrasound-based measurements such as the amniotic fluid index (AFI) or the single deepest pocket is generally considered low, discrepancies may still arise depending on factors such as ultrasound technique, fetal position, and operator expertise [35]. Furthermore, the measurement of the fore-bag volume is not routinely standardized across clinical settings, thereby limiting its generalizability. These issues highlight the need for refined protocols and potential automation to enhance consistency and clinical applicability when incorporating sonographic assessments into predictive models for ECV.

Previous studies have also focused on non-sonographic methods to assess both the likelihood of a successful ECV and the progression of a vaginal breech birth. The problem is that all these methods involve some degree of subjectivity. Thus, in the case of evaluating ECV, clinical parameters such as abdominal wall tone, uterine relaxation, maternal discomfort, and palpability of the fetal head during the maneuver are frequently considered [36]. Multiparity has consistently been shown to be a strong predictor of ECV success, as prior births often result in a more distensible uterus and compliant abdominal wall, facilitating the maneuver. Moreover, the absence of maternal pain during ECV has been independently associated with significantly higher success rates (one study found that women reporting no pain had up to 14 times higher odds of a successful version) [37]. In respect to the palpability of the fetal head, in a retrospective study by Correia Costa et al., involving 324 ECVs, a non-palpable fetal head was identified as an independent predictor of failure, with an odds ratio of 0.20 (95% CI: 0.06–0.60) [38].

Similarly, during labor progression in breech presentation, several non-sonographic clinical findings are used, such as cervical dilatation, fetal station, pelvic adequacy, and type of breech presentation, but these too are prone to observer variability. For instance, interobserver agreement in estimating fetal station is notoriously poor, and clinical pelvimetry remains subjective with limited reproducibility and uncertain predictive value [8,15,39]. Although these parameters are still considered to be essential for intrapartum decision-making, particularly in determining whether to continue with a vaginal breech birth or proceed to cesarean section, their inherent variability underscores the need for structured protocols and objective methods to enhance reliability, such as BPA measurements.

### 4.3. Strengths and Limitations

The main strength of our study lies in its multicenter and international nature, incorporating the expertise of eight obstetricians from four distinct centers across two different countries. This diversity enhances the external validity and generalizability of our results. Additionally, by having all participants analyze the same pre-acquired US volumes, we eliminated the variability in image acquisition and focused solely on interpretation, providing a strong rationale for the need to unify the measurement criteria. This process ultimately informed the development of the image-based checklist.

One limitation is that all BPA measurements were performed offline, unlike the real-time evaluations typical in clinical practice. While this may create minor discrepancies with live assessments, we argue that offline analysis offers a controlled environment for the development and refinement of standardized criteria without the external pressure characteristic of the labor ward. Notably, we have previously demonstrated that real-time assessments can achieve comparable accuracy, even among operators without prior US experience, when supported by clear criteria and appropriate training [14]. Therefore, the use of offline measurements is unlikely to have significantly affected the results. Another limitation is that all US volumes were acquired by a single expert obstetrician and also analyzed by senior clinicians, which likely resulted in a higher image quality and more accurate measurements than might be expected from less experienced operators. However, previous studies have shown that clinicians without prior US experience can be effectively trained in intrapartum ultrasound and reach performance levels similar to those of expert practitioners [14]. In addition to homogenizing the level of expertise among operators, we aimed to ensure that the results primarily reflect the impact of the intervention itself.

#### Clinical Implications

This study highlights the importance of standardizing criteria when introducing new diagnostic tools, such as intrapartum US in clinical practice. The BPA, when measured using well-defined criteria, may serve as a valuable parameter in predicting the outcomes of ECV and vaginal breech delivery. By demonstrating reproducibility among multiple experienced clinicians, our findings support the integration of BPA into routine obstetric evaluation, thereby enhancing the reliability of decision-making during labor.

Recent evidence highlights the role of ECV in reducing cesarean rates in breech pregnancies. A systematic review by Devold Pay et al. confirmed that attempting ECV at or near term significantly reduces the likelihood of cesarean delivery, affirming its value as a first-line intervention recommended by major international guidelines [6]. The effectiveness of ECV, however, hinges on the accurate ultrasonographic assessment of fetal positioning and descent. The present study strengthens the case for BPA as a clinically relevant and reproducible metric to guide such assessments. By offering objective benchmarks for breech engagement, BPA may improve the precision of decision-making by providing more accurate odds of success to the patient, increase rates of successful vaginal delivery by appropriate selection of cases undergoing ECV, and ultimately reduce unnecessary cesarean sections. Therefore, following this study, further research should therefore explore the implementation of BPA measurement in predicting labor outcomes and optimizing breech delivery management.

Finally, our findings reinforce the clinical relevance of promoting standardized, objective US-based evaluation during labor not only for BPA measurement but also to reduce variability for other parameters.

## 5. Conclusions

The BPA is a feasible and reproducible US parameter. The use of an image-based checklist supports the standardization of intrapartum US measurements, enhancing their clinical applicability and potential integration into quality control programs. Further studies are warranted to assess the predictive value of BPA in the context of ECV and vaginal breech delivery, as well as to evaluate training requirements, including the time and feasibility for broader clinical implementation.

## Figures and Tables

**Figure 1 diagnostics-15-01757-f001:**
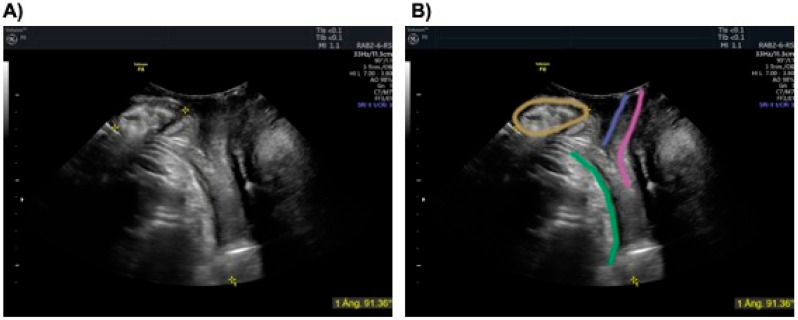
**Measurement of the breech progression angle (BPA).** Panel (**A**) shows the acquired image, which is labeled in panel (**B**). The orange line outlines the pubic symphysis (“almond” shape); the green line traces the fetal buttock; the blue line represents the urethra; and the pink line indicates the vagina. The yellow lines represent the BPA: the first line bisects the pubic symphysis down to its lower edge, and the second line extends tangentially from the lower edge of the pubic symphysis to the lowest recognizable fetal part.

**Figure 2 diagnostics-15-01757-f002:**
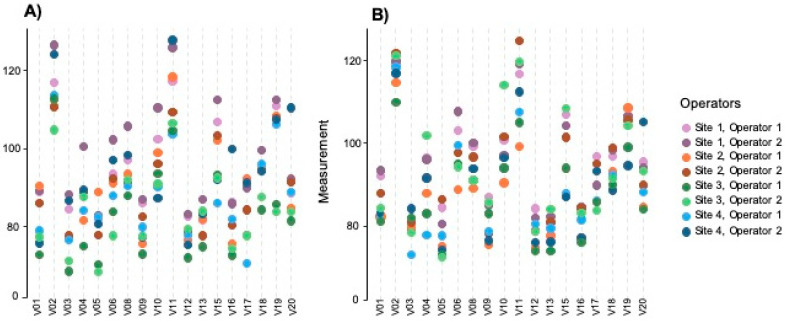
**Individual BPA measurements by each operator (colored dots) for the different volumes.** Panels show the first round of measurements before (panel (**A**)) and after (panel (**B**)) the development of the image-based checklist. The x-axis represents the evaluated volumes; volumes V07 and V14 were not analyzed due to poor image quality.

**Figure 3 diagnostics-15-01757-f003:**
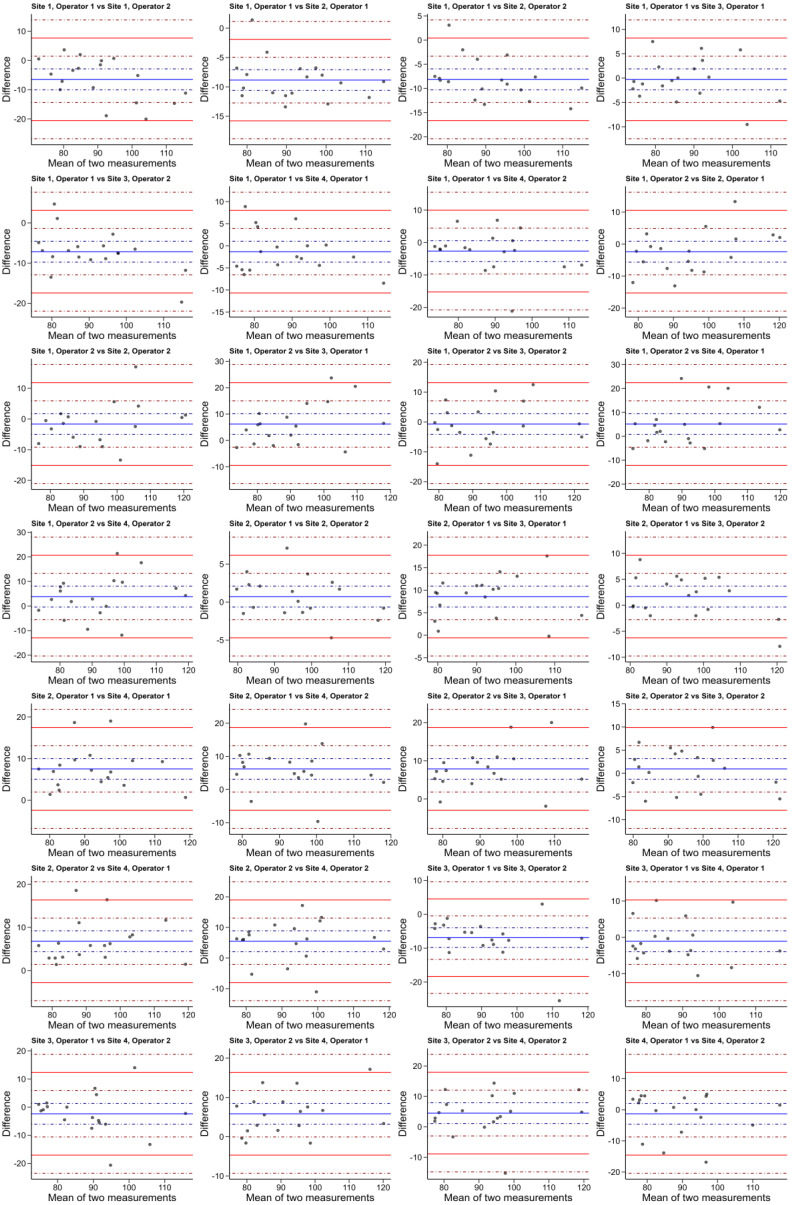
Bland–Altman plots for all pairwise comparisons of operators. Blue lines refer to mean difference, and red lines to the limits of agreement. Solid lines are point estimates and dotted lines are 95% confidence intervals.

**Table 1 diagnostics-15-01757-t001:** Image-based score for correct evaluation of breech progression angle.

Score	Image Acquisition Criteria
**1**	**Midsagittal plane**: not seeing the posterior shadow of the pubic ramus
**2**	Accurately identify the whole limits of the oval-shaped symphysis (**“almond” shape**), not just the white hyperechogenic cartilage
**3**	1st line: line along the longitudinal axis of the symphysis, dividing it into **two equal parts**
**4**	1st line: take it **to the edge of the bone**
**5**	2nd line: from the edge of the bone, **tangentially to the lowest recognizable fetal part** in the maternal pelvis

**Table 2 diagnostics-15-01757-t002:** Intraclass correlation coefficient and its 95% confidence interval for each pair of operators before (gray cells) and after (white cells) application of the image-based checklist.

	Site 1, operator 1	Site 1, operator 2	Site 2, operator 1	Site 2, operator 2	Site 3, operator 1	Site 3, operator 2	Site 4, operator 1	Site 4, operator 2
** Site 1, operator 1 **		0.89 (0.74 to 0.96)	0.61 (−0.06 to 0.89)	0.49 (−0.04 to 0.84)	0.69 (−0.06 to 0.91)	0.71 (−0.07 to 0.92)	0.61 (0.11 to 0.84)	0.54 (−0.09 to 0.84)
** Site 1, operator 2 **	0.75 (0.20 to 0.92)		0.58 (−0.10 to 0.86)	0.46 (−0.08 to 0.81)	0.63 (0.10 to 0.86)	0.69 (0.06 to 0.89)	0.57 (0.17 to 0.81)	0.49 (−0.03 to 0.78)
** Site 2, operator 1 **	0.73 (−0.06 to 0.93)	0.87 (0.68 to 0.95)		0.85 (0.22 to 0.96)	0.90 (0.68 to 0.97)	0.92 (0.39 to 0.98)	0.63 (0.26 to 0.84)	0.74 (0.46 to 0.89)
** Site 2, operator 2 **	0.75 (−0.07 to 0.94)	0.87 (0.70 to 0.95)	0.97 (0.93 to 0.99)		0.73 (−0.02 to 0.92)	0.74 (−0.07 to 0.93)	0.57 (−0.04 to 0.84)	0.73 (0.41 to 0.89)
** Site 3, operator 1 **	0.93 (0.81 to 0.97)	0.74 (0.26 to 0.91)	0.72 (−0.08 to 0.93)	0.74 (−0.05 to 0.93)		0.90 (0.77 to 0.96)	0.66 (0.31 to 0.85)	0.68 (0.36 to 0.86)
** Site 3, operator 2 **	0.77 (−0.02 to 0.94)	0.88 (0.70 to 0.95)	0.94 (0.85 to 0.98)	0.94 (0.85 to 0.98)	0.77 (0.05 to 0.93)		0.74 (0.46 to 0.89)	0.72 (0.42 to 0.88)
** Site 4, operator 1 **	0.91 (0.78 to 0.97)	0.73 (0.37 to 0.90)	0.76 (−0.05 to 0.94)	0.80 (−0.003 to 0.95)	0.88 (0.70 to 0.95)	0.83 (0.17 to 0.95)		0.63 (0.27 to 0.83)
** Site 4, operator 2 **	0.83 (0.61 to 0.93)	0.78 (0.49 to 0.91)	0.77 (0.16 to 0.93)	0.78 (0.31 to 0.92)	0.80 (0.54 to 0.92)	0.82 (0.47 to 0.93)	0.85 (0.64 to 0.94)	

**Table 3 diagnostics-15-01757-t003:** Bland–Altman statistics (mean difference, limits of agreement, and confidence intervals) for all pairs of operators.

Comparison	Mean Difference (95% CI)	Lower LoA (95% CI)	Upper LoA (95% CI)
Site 1, Operator 1 vs. Site 1, Operator 2	−6.47 (−10.05 to −2.89)	−20.59 (−26.8 to −14.39)	7.65 (1.44 to 13.85)
Site 1, Operator 1 vs. Site 2, Operator 1	−8.84 (−10.6 to −7.08)	−15.79 (−18.84 to −12.73)	−1.9 (−4.95 to 1.15)
Site 1, Operator 1 vs. Site 2, Operator 2	−8.12 (−10.28 to −5.95)	−16.66 (−20.42 to −12.91)	0.43 (−3.33 to 4.18)
Site 1, Operator 1 vs. Site 3, Operator 1	−0.26 (−2.41 to 1.89)	−8.73 (−12.45 to −5.01)	8.21 (4.49 to 11.94)
Site 1, Operator 1 vs. Site 3, Operator 2	−7.16 (−9.76 to −4.55)	−17.43 (−21.94 to −12.92)	3.11 (−1.4 to 7.63)
Site 1, Operator 1 vs. Site 4, Operator 1	−1.33 (−3.7 to 1.04)	−10.67 (−14.77 to −6.57)	8.01 (3.9 to 12.11)
Site 1, Operator 1 vs. Site 4, Operator 2	−2.63 (−5.83 to 0.56)	−15.24 (−20.77 to −9.7)	9.97 (4.43 to 15.51)
Site 1, Operator 2 vs. Site 2, Operator 1	−2.37 (−5.64 to 0.9)	−15.27 (−20.93 to −9.6)	10.53 (4.86 to 16.19)
Site 1, Operator 2 vs. Site 2, Operator 2	−1.64 (−5.06 to 1.77)	−15.09 (−21.01 to −9.18)	11.8 (5.89 to 17.72)
Site 1, Operator 2 vs. Site 3, Operator 1	6.21 (2.24 to 10.18)	−9.43 (−16.31 to −2.56)	21.86 (14.99 to 28.74)
Site 1, Operator 2 vs. Site 3, Operator 2	−0.69 (−4.2 to 2.83)	−14.53 (−20.61 to −8.44)	13.15 (7.07 to 19.24)
Site 1, Operator 2 vs. Site 4, Operator 1	5.14 (0.76 to 9.52)	−12.12 (−19.71 to −4.54)	22.41 (14.82 to 29.99)
Site 1, Operator 2 vs. Site 4, Operator 2	3.84 (−0.42 to 8.1)	−12.96 (−20.34 to −5.58)	20.64 (13.25 to 28.02)
Site 2, Operator 1 vs. Site 2, Operator 2	0.72 (−0.65 to 2.1)	−4.71 (−7.09 to −2.32)	6.16 (3.77 to 8.54)
Site 2, Operator 1 vs. Site 3, Operator 1	8.58 (6.26 to 10.91)	−0.58 (−4.61 to 3.45)	17.75 (13.72 to 21.77)
Site 2, Operator 1 vs. Site 3, Operator 2	1.68 (−0.33 to 3.7)	−6.27 (−9.76 to −2.77)	9.64 (6.14 to 13.13)
Site 2, Operator 1 vs. Site 4, Operator 1	7.51 (4.99 to 10.03)	−2.41 (−6.77 to 1.95)	17.43 (13.07 to 21.79)
Site 2, Operator 1 vs. Site 4, Operator 2	6.21 (3.05 to 9.37)	−6.25 (−11.72 to −0.77)	18.66 (13.19 to 24.13)
Site 2, Operator 2 vs. Site 3, Operator 1	7.86 (5.12 to 10.6)	−2.95 (−7.7 to 1.8)	18.67 (13.92 to 23.42)
Site 2, Operator 2 vs. Site 3, Operator 2	0.96 (−1.3 to 3.22)	−7.94 (−11.85 to −4.03)	9.86 (5.95 to 13.77)
Site 2, Operator 2 vs. Site 4, Operator 1	6.79 (4.36 to 9.21)	−2.78 (−6.99 to 1.42)	16.35 (12.15 to 20.56)
Site 2, Operator 2 vs. Site 4, Operator 2	5.48 (2.06 to 8.91)	−8.01 (−13.94 to −2.08)	18.97 (13.04 to 24.9)
Site 3, Operator 1 vs. Site 3, Operator 2	−6.9 (−9.8 to −4)	−18.34 (−23.37 to −13.31)	4.54 (−0.49 to 9.57)
Site 3, Operator 1 vs. Site 4, Operator 1	−1.07 (−3.96 to 1.81)	−12.43 (−17.43 to −7.44)	10.29 (5.3 to 15.28)
Site 3, Operator 1 vs. Site 4, Operator 2	−2.38 (−6.1 to 1.35)	−17.06 (−23.51 to −10.61)	12.31 (5.85 to 18.76)
Site 3, Operator 2 vs. Site 4, Operator 1	5.83 (3.16 to 8.49)	−4.68 (−9.3 to −0.06)	16.33 (11.72 to 20.95)
Site 3, Operator 2 vs. Site 4, Operator 2	4.52 (1.12 to 7.93)	−8.89 (−14.79 to −3)	17.94 (12.04 to 23.83)
Site 4, Operator 1 vs. Site 4, Operator 2	−1.3 (−4.67 to 2.06)	−14.57 (−20.4 to −8.74)	11.96 (6.13 to 17.79)

CI: confidence interval; LoA: limit of agreement.

## Data Availability

The original contributions presented in this study are included in the article. Further inquiries can be directed to the corresponding authors.

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
