# Peer review of "Reproducibility of Breech Progression Angle: Standardization of Transperineal Measurements and Development of Image-Based Checklist for Quality Controlâ€"

_diagnostics, 2025, doi:10.3390/diagnostics15141757_

Round 1
Reviewer 1 Report
Comments and Suggestions for Authors
I was asked to review the manuscript entitled “Reproducibility of the Breech Progression Angle: Standardization of Transperineal Measurements and Development of an Image-Based Checklist for Quality Control.”
Below, I present my comments on the manuscript:
Abstract:
The abstract formally meets the required criteria; however, based on the abstract alone, it is difficult to understand the essence of the developed scoring system and what exactly the improvement consists of. In the era of open-access publishing, the reader can, of course, access the full paper and fill in the gaps, but I wonder whether—if the word count limit allows—it would be worth expanding the abstract slightly.
Introduction:
-
“This trend has led to a decline in experience with breech births and likely to a considerable increase in maternal morbidity and mortality related to cesarean birth” – While the reader can grasp the meaning, the sentence lacks precision.
-
“such us placenta previa and accreta” – Should the authors not use the term placenta accreta spectrum instead?
-
“These methods are inherently subjective and can vary significantly between practitioners.” – This requires referencing. There is a substantial body of literature showing the inconsistency in Bishop score assessment, which is why ultrasound-based scoring systems are being developed. These include ultrasound equivalents of the Bishop score, incorporating an ultrasonographic measure of fetal station (see: https://doi.org/10.3390/jcm12134492 – this is worth mentioning, especially in the context of labor monitoring and induction).
The authors should provide a more detailed description of BPA measurements, indicating whether they apply solely to complete breech presentation or can also be performed in incomplete or footling breech presentations.
-
“extending from the most inferior portion of the pubic symphysis tangentially to the lowest recognizable fetal part in the maternal pelvis” – In cephalic presentation, this refers to a bony part; in breech presentations, could it also refer to a soft part?
Methods:
-
During the initial BPA measurements, did the physicians receive any instructions on how to perform the BPA measurement, or did they rely solely on their own experience?
-
Statistics – The authors should perform a sample size calculation.
Results:
-
Could the authors additionally include Bland–Altman plots? Visually, such plots—with marked levels of agreement—are very helpful in interpreting results.
Discussion:
I have no comments on the discussion—it is well written. However, readers might find it interesting to include an image showing the volumetric view together with the three orthogonal planes.
Congratulations on a well-prepared manuscript. I believe that after implementing my suggestions, it will be ready for publication.
Author Response
REVIEWER 1
Comment 1: Abstract: The abstract formally meets the required criteria; however, based on the abstract alone, it is difficult to understand the essence of the developed scoring system and what exactly the improvement consists of. In the era of open-access publishing, the reader can, of course, access the full paper and fill in the gaps, but I wonder whether—if the word count limit allows—it would be worth expanding the abstract slightly.
Response 1:
Thank you. The abstract has been modified as follows:
Abstract: Objectives To evaluate the reproducibility of measurements of breech progression angle (BPA) by transperineal ultrasound (US) before and after its standardization by applying an image-checklist. Methods Eighteen 3-dimensional (3D) volumes of transperineal US from women at 36-40 weeks of gestation with a singleton fetus in breech presentation were provided to eight operators from four maternity units in Spain and Italy. All operators measured the BPA using 3D US volume processing software, and inter-observer reproducibility was evaluated using the intraclass correlation coefficient (ICC). Following an online live review of all measurements by the operators, and identification of sources of disagreement, an image-based scoring system for BPA measurement was collaboratively developed. The checklist included: 1) acquisition in the midsagittal plane, avoiding the posterior shadow of the pubic ramus; 2) visualization of the complete “almond-shaped” pubic symphysis; 3) drawing a first line along the longitudinal axis of the symphysis, dividing it equally; 4) extending this line to the inferior edge of the bone; and 5) drawing a second line tangentially from the lower edge of the symphysis to the lowest recognizable fetal part. BPA measurements were then repeated using this checklist, and reproducibility was re-assessed. Results Eighteen volumes were analyzed by the eight operators, achieving a moderate reproducibility (ICC: 0.70, 95% confidence interval (CI): 0.48 to 0.86). A score was developed to include a series of landmarks for appropriate assessment of BPA. Subsequently, the same eighteen volumes were reassessed using the new score, resulting in improved reproducibility (ICC: 0.81, 95% CI: 0.66 to 0.92). Conclusions Measurement of BPA is feasible and reproducible when using a standardized image-based score.
Introduction:
Comment 2: “This trend has led to a decline in experience with breech births and likely to a considerable increase in maternal morbidity and mortality related to cesarean birth” – While the reader can grasp the meaning, the sentence lacks precision.
Response 2:
Thank you. Modified to:
“As a consequence, there has been a decline in clinical experience with breech deliveries and an associated rise in maternal morbidity and mortality related to cesarean birth ” (ref 4)
Comment 3: “such us placenta previa and accreta” – Should the authors not use the term placenta accreta spectrum instead?
Response 3:
Thank you, changed to:
“ and complications in subsequent pregnancies, such as placenta accreta spectrum”
Comment 4: “These methods are inherently subjective and can vary significantly between practitioners.” – This requires referencing. There is a substantial body of literature showing the inconsistency in Bishop score assessment, which is why ultrasound-based scoring systems are being developed. These include ultrasound equivalents of the Bishop score, incorporating an ultrasonographic measure of fetal station (see: https://doi.org/10.3390/jcm12134492 – this is worth mentioning, especially in the context of labor monitoring and induction).
Response 4:
Thank you, reference added as recommended.
Traditionally, the evaluation of fetal descent and engagement has relied heavily on digital vaginal examination or abdominal palpation. These methods are inherently subjective and can vary significantly between practitioners (I added ref (8) 10.1016/j.ejogrb.2005.04.009, Dupuis 2005)
In recent years, intrapartum ultrasound (US) has emerged as a valuable tool for providing more objective and reproducible information about fetal position, descent, and engagement (9–15) , and I added ref (16) 10.3390/jcm12134492 , Mlodawski 2023)
Comment 5: The authors should provide a more detailed description of BPA measurements, indicating whether they apply solely to complete breech presentations or can also be performed in incomplete or footling breech presentations.
Response 5:
Thank you. When the BPA was described both presentations were considered (“presentation type was complete in 10 (22.7%) cases, frank in 33 (75.0%) cases and footling in only one (2.3%) case”.The second line was described as tangentially to the lowest recognizable fetal part, whether the buttok or fetal foot. This has been clarified.
The BPA was first described by Youssef et al. in 2021, as the angle between a line running along the long axis of the pubic symphysis and another line extending from the most inferior portion of the pubic symphysis tangentially to the surface of the lowest recognizable fetal part in the maternal pelvis (whether the buttock or the lowest fetal foot), as obtained by transperineal US (21).
Comment 6: “extending from the most inferior portion of the pubic symphysis tangentially to the lowest recognizable fetal part in the maternal pelvis” – In cephalic presentation, this refers to a bony part; in breech presentations, could it also refer to a soft part?
Response 6:
Yes, thank you, we have clarified that it is the surface:
The BPA was first described by Youssef et al. in 2021, as the angle between a line running along the long axis of the pubic symphysis and another line extending from the most inferior portion of the pubic symphysis tangentially to the surface of the lowest recognizable fetal part in the maternal pelvis (whether the buttock or the lowest fetal foot), as obtained by transperineal US (21).
Methods
Comment 7: During the initial BPA measurements, did the physicians receive any instructions on how to perform the BPA measurement, or did they rely solely on their own experience?
Response 7: The operators measured the BPA based on their independent review of the original paper but without any additional training. We have clarified this in the manuscript:
To reduce the influence of baseline skill variability, operators were selected among obstetricians with expertise in obstetric US and who routinely used intrapartum US in their clinical practice, but without any previous experience measuring BPA. Although the original publication was reviewed by each operator, no specific training was provided. The operators measured the BPA using a 3D US volume processing software (4D ViewTM, version 5.0, GE Medical, Kretz Ultrasound, Zipf, Austria).
Comment 8: Statistics – The authors should perform a sample size calculation.
Response 8: Thank you, we have performed a post-hoc power analysis (explained in the Methods and presented in Results sessions).
A post hoc power analysis was conducted to verify that the study was adequately powered to assess reproducibility.
Assuming an expected ICC of 0.8 and a minimum acceptable ICC of 0.4, the post hoc power analysis indicated a statistical power of 79.5%, suggesting that the study was adequately powered to detect acceptable levels of agreement.
Results:
Comment 9: Could the authors additionally include Bland–Altman plots? Visually, such plots—with marked levels of agreement—are very helpful in interpreting results.
Response 9: Thank you. We have added Bland-Altman plots and estimates in the supplementary materials, and the appropriate information in the methods and results sections.
Methods:
Additionally, Bland–Altman plots were generated for each pair of operators to illustrate the mean difference between measurements (bias) and the limits of agreement, along with their 95% CI (23). Variability in the measurements was also visually assessed by plotting the measurements obtained for each volume.
Analyses were performed using the statistical software R (version 4.4.1) (24). The DescTools package (25) was used to calculate the ICCs, dplyr for data wrangling, table1 for the tables, BlandAltmanLeh for Bland-Altman stats and plots (26), ICC.Sample.Size for the power analysis (27), and the ggplot2 package (28) to generate the plots.
Results:
Bland-Altman plots can be seen in Figure 3 and stats in Supplementary Table 3
Discussion:
Comment 10: I have no comments on the discussion—it is well written. However, readers might find it interesting to include an image showing the volumetric view together with the three orthogonal planes.
Response 10: Thank you for your comment. We agree that the 3D volumes were valuable for conducting this reproducibility study. However, given their limited applicability in routine clinical practice within the labor ward, we chose not to include volumetric views in the manuscript.
Reviewer 2 Report
Comments and Suggestions for Authors
The idea of ​​performing transperineal ultrasound assessment is excellent, especially the attempt to find strategies to reduce cesarean sections.
I find it very interesting and praiseworthy to develop a checklist, useful for developing correct image management in the future for those who will approach these measurements, as demonstrated by the improvement of the intraclass correlation coefficient (ICC).
In fact, the checklist has proven to be more intuitive, facilitating a more confident interpretation of the anatomical landmarks.
And this aspect, in my opinion, represents a notable value for those who perform diagnostic imaging with important repercussions for clinical and prognostic purposes.
I absolutely agree that the results reinforce the importance of structured training and methodological standardization in the ultrasound assessment of these problems.
I would just be curious to know if the breech presentation variety of feet could have affected the brilliant results obtained and consequently the clinical outcome of the birth, and at what fetal anatomical level the second line should be drawn starting from the pubic symphysis (the feet or always the fetal podex?)
Author Response
REVIEWER 2:
The idea of ​​performing transperineal ultrasound assessment is excellent, especially the attempt to find strategies to reduce cesarean sections.
I find it very interesting and praiseworthy to develop a checklist, useful for developing correct image management in the future for those who will approach these measurements, as demonstrated by the improvement of the intraclass correlation coefficient (ICC).
In fact, the checklist has proven to be more intuitive, facilitating a more confident interpretation of the anatomical landmarks.
And this aspect, in my opinion, represents a notable value for those who perform diagnostic imaging with important repercussions for clinical and prognostic purposes.
I absolutely agree that the results reinforce the importance of structured training and methodological standardization in the ultrasound assessment of these problems.
Comment 1: I would just be curious to know if the breech presentation variety of feet could have affected the brilliant results obtained and consequently the clinical outcome of the birth, and at what fetal anatomical level the second line should be drawn starting from the pubic symphysis (the feet or always the fetal podex?)
Response 1: Thank you very much for your kind words and for appreciating our work. In both our bank of volumes and in those used volumes used byYoussef et al. in the original description of the BPA, there were cases of frank, complete, and footling breech presentations (see comment 5, reviewer 1). The second line should be drawn tangentially to the lowest recognizable fetal part in the maternal pelvis—whether that is the buttock or the lowest foot—depending on the presentation. This has been clarified in the manuscript, thank you.
Reviewer 3 Report
Comments and Suggestions for Authors
It is a study that makes an important contribution to assisting a natural breech birth. The proposed parameter, BPA, proves to be reproducible, with a decrease in interobserver variability, if the proposed checklist is used. But there are also limitations:
All volumes were acquired by a single operator, which may mask the real clinical variability; the operator's experience also matters; the time required for training, the duration of the operator's time should also be mentioned; additional studies are needed to validate the real predictive value.
Author Response
REVIEWER 3:
Comment 1: It is a study that makes an important contribution to assisting a natural breech birth. The proposed parameter, BPA, proves to be reproducible, with a decrease in interobserver variability, if the proposed checklist is used. But there are also limitations: All volumes were acquired by a single operator, which may mask the real clinical variability; the operator's experience also matters; the time required for training, the duration of the operator's time should also be mentioned; additional studies are needed to validate the real predictive value.
Response 1: Thank you for your comment. We have addressed these limitations in the relevant section of the manuscript, expanding on the limited impact we anticipate them to have on the results. Additionally, we have updated the conclusions to emphasize the necessity of future studies to validate the predictive value of BPA in clinical practice and to explore training requirements for broader implementation.
Discussion:
Notably, we have previously demonstrated that real-time assessments can achieve comparable accuracy, even among operators without prior US experience, when supported by clear criteria and appropriate training (14).
Conclusion:
The use of an image-based checklist supports the standardization of intrapartum US measurements, enhancing their clinical applicability and potential integration into quality control programs. Further studies are warranted to assess the predictive value of BPA in the context of ECV and vaginal breech delivery, as well as to evaluate training requirements, including the time and feasibility for broader clinical implementation.